# Transurethral Incisions for Bladder Neck Contracture: Comparable Results without Intralesional Injections

**DOI:** 10.3390/jcm11154355

**Published:** 2022-07-27

**Authors:** Samantha W. Nealon, Raj R. Bhanvadia, Shervin Badkhshan, Sarah C. Sanders, Steven J. Hudak, Allen F. Morey

**Affiliations:** Department of Urology, University of Texas Southwestern Medical Center, Dallas, TX 75390, USA; samantha.wardnealon@utsouthwestern.edu (S.W.N.); raj.bhanvadia@phhs.org (R.R.B.); shervin.badkhshan@utsouthwestern.edu (S.B.); sarah.sanders@utsouthwestern.edu (S.C.S.); steven.hudak@utsouthwestern.edu (S.J.H.)

**Keywords:** transurethral incision of bladder neck contracture, balloon dilation, adjunctive intralesional injections, artificial urinary sphincter

## Abstract

To present our 12-year experience using an endoscopic approach to manage bladder neck contracture (BNC) without adjunctive intralesional agents and compare it to published series not incorporating them, we retrospectively reviewed 123 patients treated for BNC from 2008 to 2020. All underwent 24 Fr balloon dilation followed by transurethral incision of BNC (TUIBNC) with deep incisions at 3 and 9 o’clock using a Collins knife without the use of intralesional injections. Success was defined as a patent bladder neck and 16 Fr cystoscope passage into the bladder two months later. Most with recurrent BNC underwent repeat TUIBNC. Success rates, demographics, and BNC characteristics were analyzed. The etiology of BNC in our cohort was most commonly radical prostatectomy with or without radiation (36/123, 29.3%, 40/123, 32.5%). Some had BNC treatment prior to referral (30/123, 24.4%). At 12-month follow-up, bladder neck patency was observed in 101/123 (82.1%) after one TUIBNC. An additional 15 patients (116/123, 94.3%) had success after two TUIBNCs. On univariate and multivariate analyses, ≥2 endoscopic treatments was the only factor associated with failure. TUIBNC via balloon dilation and deep bilateral incisions without the use of adjunctive intralesional injections has a high patency rate. History of two or more prior endoscopic procedures is associated with failure.

## 1. Introduction

Bladder neck contracture (BNC) is a complication of prostate surgery or radiation with incidence ranging from 0.48 to 32% [1]. Management strategies for BNC are broad, including self-catheterization, various endoscopic techniques, and reconstruction for refractory contractures [1,2,3]. Adjunctive steroid or chemotherapeutic agent injections into the treated BNC have been employed to reduce recurrence, with varying levels of success. The role of these agents remains controversial [1,4,5,6,7,8,9,10].

We present our 12-year experience on the surgical management of both de novo and recurrent BNC using a standardized endoscopic approach with combination of balloon dilation and transurethral incision, without the use of adjunctive intralesional agents. We hypothesized that our technique resulted in non-inferior surgical success rates compared to other published series. Our objective was to determine our institutional surgical success rate and risk factors for recurrent stenosis following BNC treatment. We then compared our success rates to those of previous studies using adjunctive intralesional agents.

## 2. Materials and Methods

Institutional Review Board approval was obtained for this study. We performed a retrospective review of the large reconstructive urology database at our tertiary referral center from 2008 to 2020. A total of 123 consecutive men were identified with a history of BNC due to prostate interventions who underwent transurethral incision of BNC (TUIBNC) using a standardized management protocol by a single surgeon.

### Procedural Technique

Under general anesthesia, the BNC was cannulated with a guidewire and then dilated with a 24 Fr × 4 cm Uromax balloon dilator (Boston Scientific Corp., Marlborough, MA, USA; Medline Industries, Northfield, IL, USA), which was inflated for approximately three minutes. The balloon was deflated and removed. Next, a 24 Fr cystoresectoscope was passed to the BNC, where electrocautery via Collins knife was used to incise the bladder neck at the 3 and 9 o’clock positions, using a cutting current of 30–50 W, as previously described [11]. Deep incisions were made, through the circular muscle fibers of the bladder neck and into the perivesical fat, until the resectoscope passed resistance-free into the bladder. Lateral incisions were made to avoid injury to the rectum. There was no use of adjunctive intralesional injections into the bladder neck. All patients were discharged with an indwelling 20 Fr urethral catheter for approximately four days postoperatively.

Follow-up consisted of office flexible cystourethroscopy performed by the senior author approximately two months after the original procedure to evaluate bladder neck patency and stability. Surgical success was defined as the ability to pass the 16 Fr flexible cystoscope through the bladder neck without resistance. Patients with recurrent BNC were counseled for repeat TUIBNC using the same technique. Patients were divided into two groups based on success or need for repeat intervention (1 TUIBNC vs. ≥2 TUIBNC). Data were analyzed, including age, race, body mass index (BMI), etiology of BNC, history of diabetes, smoking, prior radiation, and number of previous endoscopic BNC procedures.

Additionally, the available literature was reviewed for studies performing endoscopic management of BNC/vesicourethral anastomotic stenosis (VUAS) followed by injection of an adjunctive intralesional agent into the bladder neck scar. Success rates in these studies were reviewed and compared to one another and to the current series.

## 3. Results

A total of 123 patients underwent TUIBNC, with a median follow-up of 12 months. Mean patient age was 69 (range, 49–86), and mean BMI (kg/m^2^) was 28.4 (range, 21.6–38.4). Bladder neck contracture etiologies varied, with 40 (35.5%) of BNCs occurring after radical prostatectomy (RP), 36 (29.3%) after combination of RP with radiation, 27 (22.3%) after transurethral procedure for prostate enlargement, 8 (6.6%) after either external beam radiation (EBRT) and/or brachytherapy alone, and 9 (7.4%) following combination transurethral prostate procedure with radiation. Thirty patients (30/123, 24.4%) had one (22/30) or several (8/30) previous endoscopic procedures at other institutions prior to presenting with recurrent BNC.

Median time from TUIBNC to follow-up cystoscopy was two months. All patients in the study had a follow-up cystoscopy. Many achieved surgical success following one procedure (101/123, 82.1%). Of the 22 with recurrent BNC, 19 underwent repeat TUIBNC with the same technique. After the second TUIBNC, 15/19 (78.9%) achieved surgical success. Overall surgical success after two TUIBNC procedures was high, 116/123 (94.3%). Of the seven patients who failed TUIBNC, three were managed with chronic suprapubic tube placement, two underwent bladder neck reconstruction, and two had a third TUIBNC procedure—both with success. On univariate and multivariate analyses, the only factor associated with TUIBNC failure was ≥2 previous endoscopic treatments (Table 1 and Table 2). History of prior radiation, RP, diabetes, hypertension, and >10 pack year smoking history were not associated with TUIBNC failure.

Surgical success rates were examined among the available studies in the literature (Table 3). Studies were then grouped by intralesional injection agent and overall, with success rates compared to the current series. Percent values were calculated as the number of patients with success per total number of patients in a category. When grouped by intralesional injection agent, triamcinolone success rates were 80.5% and 83.3% after procedures 1 and 2; mitomycin C success rates were 65.3%, 80.0%, and 85.4% after procedures 1, 2, and 3 (Table 4). Overall success rates for BNC treatment after procedures 1, 2, and 3 were 68.4%, 80.3%, and 85.4% vs. 82.1% and 94.3% for the current series.

## 4. Discussion

Using the described standardized approach, we observed surgical success rates of 82.1% and 94.3% after first and second TUIBNC without any adjunctive intralesional therapies. These are high success rates, without the increased risk of adverse events or the added expense of additional agents. Our technique of initial balloon dilation under direct vision, as previously described, creates a safe tract to then atraumatically pass the resectoscope, empty the bladder, and evaluate the bladder and ureteral orifices prior to TUIBNC [16]. For many anterior urethral strictures, this initial dilation may be all that is needed to maintain patency. However, in the case of dense fibrotic BNCs, the additional adequate deep bilateral incisions are the key in de-obstructing the bladder neck fully and maintaining patency more reliably.

### 4.1. Adjunctive Intralesional Therapies and Bladder Neck Contracture Management

Triamcinolone and mitomycin C (MMC) are the most reported intralesional injection agents described for use in the treatment of BNC/VUAS (Table 3). Triamcinolone is a corticosteroid commonly used to reduce hypertrophic scars and keloids [14]. It has been injected endoscopically into BNCs in an effort to weaken contracture and theoretically prevent recurrence. Several studies have shown success with triamcinolone injections after the surgical opening of BNC, ranging from 70 to 92.9% [1,2,17]. Notably, in the group with 92.9% patency, this was maintained by office cystoscopy every four weeks with repeated triamcinolone injections into the bladder neck [17]. This technique of repeated intervention makes it difficult to compare to other studies of its kind.

Mitomycin C is a chemotherapy that cross-links DNA, leading to cell death and decreasing fibrosis and scar formation [12]. Mitomycin C injection after bladder neck incision has proven successful in 65.3%, 80.0%, and 85.4% on average after one, two, and three treatments (Table 4), which is still lower overall success than that achieved with Collins knife incision alone [4,5,6,7,8,11,15]. Although we did not find any significant association between radiation history and BNC recurrence in our study, others have reported lower surgical success in radiated patients despite MMC use. The 90% success rate reported in one MMC study was achieved in non-radiated patients, with radiated patients only having 75% success, and notably all patients in this study were being treated for recurrent BNC [8]. Another recent study using MMC injection after bladder neck incision showed that surgical success was only achieved in 45%, 69%, 78%, and 84% after one and up to four repeat procedures. Those who underwent a combination of RP and radiation experienced poorer outcomes compared to those who underwent RP or radiation alone. This patient group had a significantly shorter time to BNC recurrence [4].

Mitomycin C injection has been associated with a 7% risk of serious adverse events, including the development of osteitis pubis, bladder neck necrosis, rectourethral fistula, and rapid BNC recurrence—many ultimately needing cystectomy and permanent urinary diversion. These ramifications were associated with higher MMC doses (2.0–4.5 mg) [6]. The treatment cost of these potential adverse events is an important factor to consider.

Data examining the cost of adjunctive agents are sparse. Mitomycin C cost has been reported at ~455 USD/procedure for use in laryngotracheal stenosis, when given at comparable doses to those used for BNC [9]. The efficacy of triamcinolone as a low-cost option for the treatment of fibrosis has been reported across numerous specialties, but few report the specifics of the pricing for the medication. For intraarticular injection in an orthopedic study, 2 mL of triamcinolone cost 14.94 USD/procedure [10]. These medications are by no means prohibitively expensive, but we were able to achieve non-inferior results with our technique, without the added cost, risk, and time.

### 4.2. Management of Bladder Neck Contracture

Management techniques for BNC are broad and have been extensively described in the literature. Some strategies include self-dilation with intermittent catheterization, transurethral bladder neck resection, incision with cold knife, Collins knife, Holmium laser ablation, with and without adjunctive intralesional agent injection, urethral stenting, and open/robotic abdominal reconstruction [1,3,5,18]. Intermittent self-catheterization has been proposed for post-prostatectomy VUAS with up to 73% success after 1–2 dilations [19]. In our experience, self-dilation has poor long-term success and is not well tolerated. Because patients performing self-catheterization have been shown to have poor quality of life, this management strategy is avoided at our institution [20].

Holmium laser incision of BNC (3 and 9 o’clock or 3, 6, 9, and 12 o’clock positions) followed by injection of triamcinolone in the incision sites had 83% success in two series of VUAS [1,2]. The superior method for BNC incision, with either electrocautery (hot knife/Collins knife) or cold knife, is subject to ongoing debate. Although there are no prospective studies with the primary objective of directly comparing these two techniques, electrocautery has demonstrated superiority over cold knife incision in maintaining bladder neck patency retrospectively (63 vs. 50%, *p* = 0.03) [6]. Both techniques have demonstrated high success rates (see Table 3). Another treatment alternative to consider in this population is robotic YV-plasty, especially in those with recalcitrant BNC after repeated attempts at endoscopic management. Two published series report surgical success rates of 83.3–100%, with minimal patient morbidity [21,22].

Endoscopic techniques are difficult to compare due to small patient series with diverse etiologies and stages of treatment, non-standardized, heterogenous approaches with varied methods, and short follow-up duration. Table 4 details the surgical success rates of available studies based on the adjunctive intralesional agent utilized. Those injected with triamcinolone had higher surgical success compared to those injected with MMC; however, there were far fewer patients in the triamcinolone group. The current series’ success rates were higher than with either intralesional injection agent (Table 4). When grouped by surgical technique or when all studies were grouped together overall, the current series using Collins knife alone had the highest surgical success rates after one and two procedures.

### 4.3. Pathophysiology of Bladder Neck Contracture

The exact pathophysiology for the development of BNC is poorly understood. Associations have been made to surgical approach, surgeon, treatment, and specific patient, anatomical, and voiding characteristics [23,24]. It has also been proposed that the higher prevalence of peripheral vascular disease in older patients may be related to an increased risk of BNC [23]. For those with recalcitrant BNC after previous endoscopic procedures, frozen periurethral tissue is likely the culprit. Previous incisions/resections were not quite deep enough to fully release the dense contracture.

Established comorbid conditions associated with BNC include coronary artery disease, hypertension, diabetes mellitus, and smoking, which all affect the microvasculature, leading to poor wound healing and scarring. Smoking is a well-documented deterrent of wound healing, especially at the time of prostatectomy [25]. Wounds in smokers have vasoconstriction, poor collagen synthesis, and vascular endothelial dysfunction [26]. When compared to non-smokers, smokers have enhanced wound contraction, thought to be due to the increased presence of myofibroblasts [26]. In our current larger series, only those with a history of ≥2 endoscopic procedures had a statistically significant higher risk of BNC recurrence compared to our previous BNC series, which showed higher BNC recurrence both with >2 previous endoscopic procedures as well as >10 pack year smoking history [18]. Regardless, smoking cessation in these patients should still be highly encouraged because of the well-established general health benefits and for minimizing wound contraction postoperatively.

It is worth noting that transurethral procedures for the treatment of benign prostatic enlargement were the etiology for over 22% of BNC cases. Transurethral resection of the prostate (TURP) leads to BNC development in 0.4–9.7% of cases [27,28]. There is a higher likelihood of BNC in those with small prostates—low adenoma weight or <30 g of adenoma resection [27,28]. Patients undergoing TURP with concomitant prophylactic transurethral incision of the bladder neck are less likely to develop BNC than those who do not [24,27]. Bladder neck contracture incidence after transurethral procedures using various laser energies has also been documented: 0–5.9% for GreenLight, 1.6–3.6% for Thulium, and 1.1–9.6% for Holmium lasers, respectively [28]. As these techniques and energies are employed more frequently in both community and academic practice, we may encounter more BNCs.

### 4.4. Limitations

This study was retrospective and was performed at a high-volume tertiary referral center by a single surgeon. Our results are therefore susceptible to selection bias as many patients have been specifically referred to us after failed attempts by other surgeons. Gathering data regarding previous surgical history can also be difficult. We often see limited follow-up duration as many of our patients are referred from out of state, with long-term follow-up continuing with their outside urologists. We do not routinely perform cystoscopy past two months in these patients unless obstructive symptoms recur/develop. Future studies would be improved with confirmatory cystoscopies at six months and one year. Our early success may not necessarily translate into long-term success.

Additionally, the overall level of evidence for any BNC treatments is low since most studies are small retrospective cohort or case–control studies. Study comparisons are difficult because there is no standardized grading scale for BNC severity. Prospective multi-center randomized control trials are needed comparing BNC management techniques with and without adjunctive therapies and with longer follow-up to fully elucidate treatment benefits.

## 5. Conclusions

Balloon dilation followed by TUIBNC via deep bilateral incisions has an 82% and 94% patency rate after one and two procedures without the use of adjunctive intralesional injections in the bladder neck. As the use of adjunctive agents gains popularity, our cohort results add to the existing body of literature as a baseline upon which to compare future outcomes. History of two or more prior endoscopic procedures is associated with TUIBNC failure.

## Figures and Tables

**Table 1 jcm-11-04355-t001:** Univariate analysis of TUIBNC success vs. failures.

Variable	Success	Failure	*p*-Value
**Age**			0.13
Mean (SD)	70 (8.0)	67 (8.0)	
Median (IQR)	70 (65–76)	68 (63–72)	
**Race, *n* (%)**			0.4
White	73 (72.3)	14 (63.6)	
Other	28 (27.7)	8 (36.4)	
**BMI, *n* (%)**			0.6
<30	65 (64.4)	13 (59.1)	
≥30	36 (35.6)	9 (40.9)	
**Diabetes History, *n* (%)**			0.2
No	75 (74.3)	19 (86.4)	
Yes	26 (25.7)	3 (13.6)	
**Radiation History, *n* (%)**			0.8
No	56 (55.5)	13 (59.1)	
Yes	45 (44.6)	9 (40.9)	
**>10 Pack Year Smoking, *n* (%)**			0.5
No	54 (58.7)	14 (66.7)	
Yes	38 (41.3)	7 (33.3)	
**Previous TUIBNC, *n* (%)**			**<0.01**
<2	98 (97.0)	17 (77.3)	
≥2	**3 (3.0)**	**5 (22.7)**	
**Radical Prostatectomy**			0.16
No	44 (43.6)	6 (27.3)	
Yes	57 (56.4)	16 (72.7)	

SD—standard deviation, IQR—interquartile range, TUIBNC—transurethral incision of bladder neck contracture, BMI—body mass index.

**Table 2 jcm-11-04355-t002:** Prediction of TUIBNC failure—multivariable logistic regression (N = 113).

Variable	Odds Ratio	*p*-Value	95% Confidence Interval	
**Age**	0.97	0.42	0.91	1.04
**Race**				
White	Ref.	Ref.	Ref.	Ref.
Other	2.03	0.20	0.68	6.06
**BMI**				
<30	Ref.	Ref.	Ref.	Ref.
≥30	1.33	0.62	0.42	4.18
**History of Diabetes**	0.60	0.50	0.14	2.59
**History of Radiation**	1.44	0.52	0.47	4.46
**More than 10 Pack Year Smoking History**	0.72	0.59	0.23	2.31
**More than 2 Prior TUIBNC**	**12.97**	**0.01**	1.85	91.03
**Prior Radical Prostatectomy**	1.61	0.45	0.47	5.50

TUIBNC—transurethral incision of bladder neck contracture, BMI—body mass index.

**Table 3 jcm-11-04355-t003:** Summary of studies utilizing endoscopic management of BNC/VUAS with adjunctive therapies.

Publication (Year)	Etiology of BNC/VUAS	N	Treatment of BNC	Dose	Mean (*)/Median (#) Follow-Up	Success Rates: Cystoscopic Documentation of Stable Bladder Neck
Eltahawy (2008) [2]	RP	24	-Holmium laser incision at 3 and 9 o’clock -Injection of **triamcinolone**	80 mg	24 mo (*)	19/24 (83%)
Vanni (2011) [7]	RP, radiation, TURP	18	-Tri- or quadrant CKI of bladder neck -Injection of **MMC** at each incision site	0.3–0.4 mg/mL	12 mo (#)	-1 procedure
						13/18 (72%)
						-2 procedures
						16/18 (89%)
Redshaw (2015) [6]	RP, radical cystectomy w/neobladder, radiation, simple prostatectomy, TURP, pelvic fracture	55	-3 or 4 deep bladder neck incisions until fat seen; either CKI or Collins knife	Range: dose from 0.4 to 10 mg	9.2 mo (#)	-1 procedure 32/55 (58%)
			-MMC injection into wound bed	Concentration from 0.1 to 1.0 mg/mL		-2 procedures 41/55 (75%)
Farrell (2015) [5]	Radiation- and non-radiation-induced strictures	37	-CKI at 3, 6, 9, 12 o’clock followed by MMC injections	4 mg/	23 mo (#)	28/37 (75.7%)
			-CIC once daily to maintain patency	10 mL		
Nagpal (2015) [12]	Highly recurrent BNC (≥1 prior incision procedure)	40	-CKI of bladder neck followed by **MMC** injections at each incision site	0.3–0.4 mg/mL	20.5 mo (*)	-1 procedure
						30/40 (75%)
						-2 procedures
						35/40 (87.5%)
Sourial (2017) [13]	RP for localized prostate cancer	29	-**MMC** injected at 3, 6, and 9 o’clock -Serial urethral dilation to 26 Fr	0.05 mg/mL	12 mo (#)	-1 procedure
						23/29 (79.3%)
						-2 procedures
						25/29 (86.2%)
Zhang (2020) [14]	Highly recurrent BNC (≥2 prior procedures) after TURP	28	-TUR bladder neck at 2–3 o’clock followed by **triamcinolone** injection at 3, 6, 9, 12 o’clock -Office cysto with triamcinolone every 4 weeks × 3	80 mg	33.6 mo (#)	26/28 (92.9%)
Mann (2021) [1]	RP, +/− radiation	30	-**Triamcinolone** injection into scar at 3, 6, 9, 12 o’clock -Holmium laser ablation at injection sites	40 mg/2 mL	33.3 mo (*)	-1 procedure 21/30 (70%)
						-2 procedures
						25/30 (83.3%)
Rozanski (2021) [8]	RP, benign prostate surgery, radiation Recurrent BNC after failed DVIU or catheterization	86	-CKI at 3 and 9 o’clock if EUS involved or CKI at 3, 6, 9, 12 o’clock if EUS not involved -**MMC** injected at each incision site	0.3–0.4 mg/mL	21 mo (#)	-1 procedure 56/86 (65%)
						-2 procedures 71/86 (83%)
						-3 procedures 77/86 (90%)
Selvaraj (2021) [15]	TURP Recurrent BNC after ≥3 prior procedures	10	-TUR bladder neck -10-point **MMC** injection in resection site	2 mg	24 mo (*)	8/10 (80%)
Hacker (2022) [4]	RP, EBRT, or RP-EBRT	51	-Plasma cut at 3 and 9 o’clock -**MMC** injection to incisions	2 mg/5 mL	32 mo (#)	-1 procedure 23/51 (45%)
						-2 procedures 35/51 (69%)
						-3 procedures 40/51 (78%)
						-4 procedures 43/51 (84%)
**Current Series**	**Varied etiologies**	**123**	**Deep bilateral bladder neck incisions with Collins knife at 3 and 9 o’clock**	**NA**	**12 mo (#)**	**-1 procedure 101/123 (82.1%)**
						**-2 procedures 116/123 (94.2%)**

BNC—bladder neck contracture, CKI—cold knife incision, MMC—mitomycin C, CIC—clean intermittent catheterization, TURP—transurethral resection of prostate, DVIU—direct vision internal urethrotomy, EUS—external urinary sphincter, VUAS—vesicourethral anastomotic stenosis, RP—radical prostatectomy, EBRT—external beam radiation therapy, *—mean, #—median.

**Table 4 jcm-11-04355-t004:** Average surgical success rates by intralesional injection agent.

Intralesional Injection Agent	Procedure 1	Procedure 2	Procedure 3
**Triamcinolone**			
Eltahawy [2]	19/24	Not Reported	Not Reported
Zhang [14]	* 26/28	Not Reported	Not Reported
Mann [1]	21/30	25/30	Not Reported
**Totals (%)**	**66/82 (80.5)**	**25/30 (83.3)**	---
**Mitomycin C**			
Vanni [7]	13/18	16/18	Not Reported
Redshaw [6]	32/55	41/55	Not Reported
Farrell [5]	28/37	Not Reported	Not Reported
Nagpal [12]	30/40	35/40	Not Reported
Sourial [13]	23/29	25/29	Not Reported
Rozanski [8]	56/86	71/86	77/86
Selvaraj [15]	8/10	Not Reported	Not Reported
Hacker [4]	*ε* 23/51	35/51	40/51
**Totals (%)**	**213/326 (65.3)**	**223/279 (80.0)**	**117/137 (85.4)**
**Current Series**	101/123	116/123	Not reported
**Totals (%)**	**101/123 (82.1)**	**116/123 (94.3)**	---

Values are reported as number of patients per total number. * Study protocol of scheduled repeat cystoscopies and injection of triamcinolone at regular intervals. *ε* Procedure 4 results not included—only study to include data from 4 separate procedures.

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
