# Peer review of "Transurethral Incisions for Bladder Neck Contracture: Comparable Results without Intralesional Injections"

_jcm, 2022, doi:10.3390/jcm11154355_

Round 1
Reviewer 1 Report
The authors present an impressive series of 123 patients treated over a 12-year period with a relevant follow-up period. The methodology used in the study is appropriate and provides clear information and makes easy reading and interpretation. I also concur with the authors that adjunct antifibrotic injections may not be necessary, and even deleterious as they are not exempt of complications, besides intrinsic additional costs!
I congratulate the authors for this very useful study.
The only snag I would point out is for how long these incised VUASs will remain patent?!
Author Response
We appreciate reviewer 1's comments on our study. Please see the below attachment to view changes made to the manuscript and notes to reviewer 2.

Reviewer 2 Report
It is a comparably large cohort of a very special patient population. The intervention is interesting and impressive. However, the study lacks clear outcome definitions and clinical parameters. The success is not clearly defined, ultimately only a urethroscopy after 2 months is described. The 12-month follow-up is not defined in more detail. Clinical data (uroflow, sonography, PROMS, QoL) and information on side effects are completely lacking. If these were added, it would be a great paper
Abstract
13 success rate was defined 2 month after surgery, at line 17 only the results after 12 month were mentioned
15 please add “in our cohort”
18 additional means all 22 patients without BN-patency after 1 year were treated and 15 had success. Then perhaps it would be better to write 15 of 22 patients with 2nd treatment….. How long was the follow up of these 2nd treatment?
Methods
53 why 3 nd 9 o clock, are there references for this technique or is it a new or modified aproach?
58 is there an evidence for catheter-length?
63 discutable indication for revision. 15 Ch passage and complete bladder emptying seems to be adequate.
Results
74 as named above, follow up is a little bit misunderstoodable. 2 month cystoscopy. Means mean 12 months follow up that a second cystoscopy has been performed? Or was the “long” follow up only clinically (uroflowmetry, residual urine?) This should be named clearly!
76 absolute numbers of most reasons for BNC should be named in detail
81 discutable indication for cystoscopy
88 (Table 1) Were patients after 1st or 2nd TUIBNC-failure analyzed?
104 table 3 is a little bit chaotic
Discussion
117 adverse events were not stated in results section. What means without increased AE, increased from what?
215 as named above, absolute numbers of each reason for BNC should be stated in detail
Author Response
We appreciate reviewer 2's insight on our manuscript and have attached our specific responses in a word document.
